# Exploring the Role of Inflammation and Metabolites in Bell’s Palsy and Potential Treatment Strategies

**DOI:** 10.3390/biomedicines13040957

**Published:** 2025-04-13

**Authors:** Jiaye Lu, Ziqian Yin, Youjia Qiu, Yayi Yang, Zhouqing Chen, Jiang Wu, Zhong Wang

**Affiliations:** Department of Neurosurgery & Brain and Nerve Research Laboratory, The First Affiliated Hospital of Soochow University, 188 Shizi Street, Suzhou 215006, China; 20234232050@stu.suda.edu.cn (J.L.); sudayzq@163.com (Z.Y.); qiu_youjia@163.com (Y.Q.); yy_yang_cn@163.com (Y.Y.); wangzhong761@163.com (Z.W.)

**Keywords:** Bell’s palsy inflammation, metabolism, Mendelian randomization, bioinformatics, therapeutic strategies

## Abstract

**Introduction**: Bell’s palsy is a common acute peripheral neurological disorder causing unilateral facial paralysis. Its exact etiology remains unknown, but it is linked to inflammation, immune responses, infections, and ischemia. This study explores the potential causal relationship between Bell’s palsy and peripheral blood inflammatory proteins, metabolites, and immune cell characteristics. **Methods**: Genetic data for Bell’s palsy were obtained from the Finnish database (version R10) and IEU OpenGWAS. A two-sample Mendelian randomization (MR) approach was applied, analyzing 4907 plasma proteins, 731 immune cell traits, 91 inflammatory proteins, and 1400 metabolites. The Finnish dataset served as the discovery cohort, while the IEU OpenGWAS dataset acted as the validation cohort. Bioinformatics analyses included protein–protein interaction (PPI) networks, Gene Ontology (GO) enrichment, Kyoto Encyclopedia of Genes and Genomes (KEGG) pathway analysis, colocalization, and Linkage Disequilibrium Score Regression (LDSC) to identify candidate proteins and explore potential therapeutic targets. **Results**: MR analysis identified 70 inflammatory proteins, 77 metabolites, and 26 immune cell traits as potentially causally associated with Bell’s palsy. After external validation, BLVRB, HMOX2, TNFRSF12A, DEFB128, ITM2A, VEGF-A, and DDX58 remained significantly associated (*p* < 0.05). PPI network analysis led to 31 candidate proteins, and six core proteins (JAK2, IL27RA, OSM, CCL19, SELL, VCAM-1) were identified. **Conclusions**: Our study identifies causal relationships between inflammatory proteins, metabolites, immune cells, and Bell’s palsy, highlighting that the JAK/STAT signaling pathway may be a potentially critical target for intervention in Bell’s palsy, and that its modulation may provide new directions and opportunities for therapeutic strategies and drug discovery for the disease.

## 1. Introduction

Bell’s palsy is an acute condition characterized by paralysis of the facial nerve, with its exact etiology still undefined. During facial nerve release surgery, the surgeon often observes non-specific inflammatory manifestations of the nerves in the neural tube, such as swelling of the affected lateral nerve and signs of high pressure in the facial neural tube [1]. The symptoms of Bell’s palsy usually present as a sudden paralysis of the muscles on one side of the face, resulting in loss of frontal wrinkles, incomplete closure of the eyelids, shallow nasolabial folds, drooping of the corners of the mouth, and potentially loss of taste and hearing. The incidence of Bell’s palsy is estimated to be between 15 and 30 cases per 100,000 people per year, and although it can occur at any age, it is more prevalent in middle-aged women [2]. From the onset of a patient’s illness, his symptoms typically develop rapidly, usually within hours to days. While most patients recover spontaneously within weeks to months, a significant number of patients are still left with varying degrees of facial dysfunction after undergoing surgical procedures. There is often no effective treatment for these patients, which can have a profound impact on their daily lives and mental health [3].

Regarding the etiological mechanisms of Bell’s palsy, researchers have proposed several hypotheses, including viral infection, inflammation, autoimmune response, and ischemic damage to the facial nerve [4,5,6]. The theory of viral infection suggests that viruses such as herpes simplex virus (HSV-1) and varicella zoster virus (VZV) can induce inflammation and oedema in the facial nerve through activation or reactivation, which may lead to facial paralysis [7,8]. Gu et al. demonstrated that CD4^+^, IL-2 and IL-4 are involved in HSV-1-induced facial nerve paralysis [1]. Furthermore, related studies found that patients with Bell’s palsy had elevated concentrations of cytokines in serum samples compared to controls, including IL-1 and IL-6 [9]. In addition, in the acute phase of Bell’s palsy, a decrease in the percentage of total T cells (CD3) and T helper (CD4) was observed compared to the control group [10]. It is noteworthy that during the 24 days prior to the onset of clinical paralysis, there was a marked decrease in peripheral blood T-lymphocytes with a corresponding increase in B-lymphocytes [11].

These findings collectively suggest that peripheral blood inflammation and immune responses are strongly associated with the development of Bell’s palsy. Mendelian randomization (MR), an emerging epidemiological research method, minimizes selection-induced bias by exploiting genetic variation before exposure and outcome and obtains results that mimic those of randomized controlled trials [12,13]. MR determines causal relationships between outcomes and exposure factors by setting reasonable statistical thresholds and selecting instrumental entity variables. As a result, researchers have used MR technology to investigate risk factors for a variety of diseases, including peripheral neurological disorders, respiratory disorders and other research hotspots. By analyzing their potential relationships as well as their function, new treatment options for diseases are explored.

The present study used a two-sample MR study design, and two databases screened for to-be-selected outcomes as a discovery cohort (Finnish database) and an external validation cohort (IEU OpenGWAS). The exposure data incorporated a wide range of datasets including many kinds of factors including the plasma inflammation protein, immunoinflammation-associated proteins, immunological cellular characteristics and plasma metabolites. Our study aims to explore the effect of the core proteins in the course of the disease and the causal relationship by bioinformatics analyses, providing new treatment ideas for Bell’s palsy.

## 2. Materials and Methods

### 2.1. Research Design

To explore the causal association of serum metabolites, immune cells and related inflammatory proteins in Bell’s palsy and their role in pathogenesis, and to discover potential drug targets and biomarkers based on which we performed MR analysis and bioinformatics analysis, we used the Finnish database (version R10) of Bell’s palsy data as the initial discovery cohort and IEU Open GWAS as the validation cohort to further enhance the reliability of our results. This study strictly adhered to the STROBE-MR guidelines. A detailed overview of the study’s steps is shown in Figure 1.

### 2.2. Sources of Data

The outcome data related to Bell’s palsy were obtained from the Finnish database version R10 (https://www.finngen.fi/en/access_results) and raw data from IEU OpenGWAS. The exposure data are divided into two main categories: inflammatory and metabolic. Serum metabolite data were obtained from the GWAS catalogue, which includes 309 metabolite ratios and 1091 blood metabolites. The MR analysis regarding inflammation consisted of three components: 731 immune cells, 4907 plasma proteins and 91 inflammatory proteins. All original data used for the MR in this study were derived from individuals of European ethnicity. Exposure data for Finland can be found in Appendix A, and for IEU OpenGWAS, see Appendix A.

#### 2.2.1. Plasma Protein Screening

Plasma protein data were obtained from a GWAS study by deCODE, which included a total of 4907 proteins by protein quantitative trait loci (pQTL), with data from 35,559 Icelanders. In addition, we downloaded the human-related genomes (H, C1–C8) from the Gene Set Enrichment Analysis (GSEA) website and filtered the genes using “inflammation” and “immunity” as keywords, yielding a total of 5886 genes, as detailed in Appendix A. We cross-referenced these genes with 4907 plasma proteins and obtained a total of 925 inflammatory protein data.

#### 2.2.2. Merging with the GWAS Catalogue

Data for 91 inflammatory proteins were obtained from the GWAS catalogue and combined with 925 proteins associated with inflammation and immunity, resulting in data on 1016 proteins that were listed in the subsequent analyses.

#### 2.2.3. Selection of Immune Cells

Using the same methodology, we obtained data on 731 immune cells from the GWAS catalogue.

### 2.3. Genetic Instrumental Variable (IV) Selection

#### 2.3.1. Identification of SNPs Significantly Associated with the Phenotype

We initially identified SNP as significantly associated with phenotype using a strict threshold (*p* < 5 × 10^−6^). For serum metabolites, we first performed MR with a relatively lenient threshold (*p* < 1 × 10^−5^) and then repeated the analysis on the positive results obtained with a more stringent threshold. All the SNPs included in this study can be found in Appendix A.

#### 2.3.2. Detection of SNPs Strongly Linked to the Phenotype

We chose the more stringent criteria (LD: r^2^ < 0.001, kb > 10,000) and filtered for exposure-associated SNPs using the “clump_data” function in the “TwoSampleMR” R package (version 0.5.8).

#### 2.3.3. Integration, Concordance, and Correction of Palindromic SNPs

We integrated the concordance of exposure data and filtered SNPs with palindromic sequences based on allele frequency information using the “harmonise_data” function in the “TwoSampleMR” R package.

#### 2.3.4. Evaluation of IV Strength

We assessed the reliability of the IVs and filtered out potentially weak IVs (F < 10) by the F value. F value was calculated as F = (Beta/SE)^2^. Beta is the estimated effect size of the SNP and SE is the standard error of the effect estimate [14,15].

### 2.4. MR Analysis and Sensitivity Analysis

The results of this MR analysis were based on a random-effects model, and the causal relationship between exposure and outcome was mainly assessed using the inverse variance weighting (IVW) method. MR–Egger regression and the weighted median method were used as supplementary references, and the Wald ratio method was used to analyze exposures for which only one SNP was extracted [16,17]. To assess horizontal pleiotropy, the MR–Egger method and MRPRESSO (MR pleiotropy residual and outlier) method were jointly utilized. To guarantee the reliability and robustness of the findings, we also assessed the heterogeneity of the results using Cochran’s q-test, which indicated heterogeneity when *p* < 0.05. The Linkage Disequilibrium Score Chisq (LDSC) method is a reliable technique for genetic correlation studies in diseases or traits with intricate genetic backgrounds. It measures the genetic contribution to a trait’s variability using single nucleotide polymorphisms (SNPs) and evaluates the genetic relationship between two traits using Chi-squared statistical measures [18,19]. In our research, we utilized LDSC to investigate the genetic links between key proteins associated with Bell’s palsy.

### 2.5. Bioinformatics Analysis

As a result of these analyses, we identified a total of 70 to-be-selected positive proteins that may have a potential causal relationship with Bell’s palsy. For further analysis, bioinformatics analysis of these proteins was performed in this study.

#### 2.5.1. Protein–Protein Interaction (PPI) Online Network

First, we accessed the STRING online network to retrieve and validate the roles and mechanisms of the above 70 inflammatory immune-related proteins. Then, we constructed a more comprehensive PPI network based on the currently known interactions and functional relationships.

#### 2.5.2. Gene Ontology (GO) Enrichment Analysis and Kyoto Encyclopedia of Genes and Genomes (KEGG) Analysis

To delve deeper into the functions of these proteins in relation to disease, these 70 inflammation-related proteins were subjected to GO enrichment analysis and KEGG pathway analysis in our experiments.

#### 2.5.3. Core Protein Screening

We identified key regulatory proteins and functional modules in the PPI online network using the Molecular Complex Detection (MCODE) plugin in Cytoscape (version 3.7.1), and for optimal results, we set the parameters to node score cutoff = 0.2, degree cutoff = 2, max depth = 100 and k-core = 2.

#### 2.5.4. Colocalization Analysis

We performed further colocalization analyses of the hub proteins identified by MCODE using the ‘color’ function in the R language package. In total, there are five possible hypotheses for colocalization:

**H0:** 
*Neither related to Bell’s palsy nor inflammatory proteins (PP0).*


**H1:** 
*Related to inflammatory proteins, but not to Bell’s palsy (PP1).*


**H2:** 
*Related to Bell’s palsy, but not to inflammatory proteins (PP2).*


**H3:** 
*Related to both inflammatory proteins and Bell’s palsy, but using independent SNPs (PP3).*


**H4:** 
*Related to both inflammatory proteins and Bell’s palsy, with shared SNPs (PP4).*


## 3. Results

### 3.1. MR Analysis Results and Sensitivity Analysis

Through magnetic resonance analysis of a wide range of peripheral blood inflammatory proteins, immunological cell characteristics, and metabolites, we identified some potential positive biomarkers causally related to Bell’s palsy. In total, 70 inflammation-related proteins, 77 blood metabolites, and metabolite ratios, and 26 immune cells were classified as pending results. Raw results of MR analyses and analyses of heterogeneity and pleiotropy of Finnish data are shown in Appendix A and the IEU OpenGWAS exposure data with Bell’s palsy are shown in Appendix A.

#### 3.1.1. Screening for Inflammation-Related Proteins

From the initially screened 925 inflammation- and immunity-related proteins and 91 inflammatory proteins, MR analysis identified 70 biomarkers with causal relationships to Bell’s palsy (Appendix A). Heterogeneity and sensitivity analyses were negative among these potentially positive proteins. After external validation, we found that the *p*-values for BLVRB, HMOX2, TNFRSF12A, DEFB128, ITM2A, DDX58, and VEGF-A remained below 0.05. We categorized the potential positive results with a Beta value greater than 0 as having a positive association with Bell’s palsy, while those with a Beta value less than 0 were classified as having a negative association with Bell’s palsy.

#### 3.1.2. Immune Cells

After analyzing 731 immune cell traits and removing heterogeneity and pleiotropy, we found that 26 outcomes were causally related to Bell’s palsy (Appendix A). However, the external inflammation results for these 26 outcomes were not favorable.

#### 3.1.3. Results of Metabolites

In total, 1400 metabolites were included in the TwosampleMR analysis and subsequent elimination of heterogeneity and pleiotropy identified 60 metabolites (Appendix A) and 17 metabolite ratios (Appendix A) as potentially causally related to Bell’s palsy.

### 3.2. Results of Bioinformatics Analysis 

#### 3.2.1. PPI Network Analysis

We performed a PPI network on 70 potential positive inflammatory proteins using the STRING database, with the minimum required interaction score set to high confidence (0.700). Next, we identified 31 proteins after eliminating proteins with no correlation from the result (Figure 2A). Notably, IL-6 exhibited the highest number of connections within the interaction network (Figure 2B). This suggests that the inflammatory response is closely related to the occurrence of Bell’s palsy. Two groups of six core proteins were identified by the Cytoscape species MCODE plugin (Figure 2C,D).

#### 3.2.2. GO and KEGG Analysis

Our study performed GO and KEGG analyses of these 70 potential positive inflammatory proteins utilizing the STRING online database. After sorting the results according to −log FDR, the most important biological processes (BPs) were found to be immune system processes, whereas the cellular components (CCs) were mainly located in the extracellular region, and the molecular functions (MFs) were involved in receptor–ligand activity and signaling receptor binding. Only the top ten results were selected for BP. KEGG pathway highlighted that the most important pathway was cytokine–cytokine receptor interaction (Appendix A). Notably, our analysis highlights the crucial role these proteins play in the immune system, while also acknowledging the significant role of the inflammatory system.

#### 3.2.3. Results of Core Protein Screening

We further visualized the 26 potential results obtained from STRING using Cytoscape software. Subsequently, we employed the MCODE plugin in Cytoscape to identify core proteins. Ultimately, we identified two subnetworks composed of six hub proteins: JAK2, IL27RA, OSM, CCL19, SELL, and VCAM-1. These proteins may be more closely related to the development of disease. The main results are shown in Table 1.

#### 3.2.4. Results of Colocalization Analysis

Four proteins, VCAM-1 (PP.H4: 0.022), CCL19 (PP.H4: 0.007), OSM (PP.H4: 0.011), and IL27RA (PP.H4: 0.02), were inflamed in the colocalization analysis (Figure 3). However, it is worth noting that negative colocalization results do not necessarily indicate invalid MR results.

#### 3.2.5. Genetic Correlation Analysis

The LDSC results further validated the findings from the hub protein MR (Table 1). LDSC identified significant genetic associations between CCL19, VCAM1, OSM, and Bell’s palsy. (Genetic correlation between CCL19 and Bell’s palsy: 0.2105, *p* = 0.042; genetic correlation between VCAM1 and Bell’s palsy: 0.2478, *p* = 0.017; genetic correlation between OSM and Bell’s palsy: 0.41934, *p* = 0.021.)

## 4. Discussion

Bell palsy is an acute onset peripheral facial neuropathy and is the most common cause of lower motor neuron facial palsy, accounting for 60~75 percent of all unilateral facial palsy cases, with an annual incidence of about 1530/100,000, with a peak incidence in the 20s and 40s [1,5,20]. The clinical features of the disease include unilateral acute facial nerve palsy, often accompanied by pain behind the ear, taste disturbances, facial sensory abnormalities and auditory hypersensitivity, which severely affects the patient’s quality of life and has a significant impact on his physical and mental health and social functioning [21]. Although studies have suggested that immune abnormalities, viral infections and ischemia may mediate the pathogenesis of the disease, the exact cause of the disease has not been fully elucidated [22]. In recent years, increasing evidence has supported the key role of inflammatory immune response induced by viral infection in the development of Bell’s palsy [23,24]. Therefore, the present study was based on MR analysis to screen and identify inflammatory proteins, metabolites, and immune cells with a potential causal relationship with Bell’s palsy, and to explore the role of inflammatory proteins and immune proteins in the development of the disease by combining with bioinformatics methods.

In this study, 70 inflammation-related proteins that may be causally associated with Bell’s palsy were identified by MR analysis, and 7 significantly related proteins were screened by external dataset validation, including BLVRB, HMOX2, TNFRSF12A, DEFB128, ITM2A, DDX58 and VEGF-A. Based on GO functional enrichment analysis, these proteins were found to be mainly involved in biological processes such as immune system processes, extracellular regions, positive regulation of biostimuli and receptor–ligand activity. Further KEGG pathway analyses highlighted the potential role of cytokine–cytokine receptor interactions and rheumatoid arthritis pathways in the pathogenesis of Bell’s palsy, further supporting the association of the disease with neuroinflammation, immune imbalance and infection.

To further elucidate the potential role of inflammatory immune-related proteins in Bell palsy, we constructed a PPI network and screened 31 high-confidence interacting proteins, among which IL-6 showed a high degree of connectivity. Further analysis showed that VCAM-1, CCL19, IL27RA and IL-6 play important roles in immunomodulation, cell adhesion, inflammatory response and immune cell migration. In addition, based on the MCODE plugin, we identified six core proteins, including VCAM-1, CCL19, IL27RA, OSM, SELL and JAK2, of which VCAM-1, CCL19, OSM and IL27RA passed co-localization analysis.

Our research suggests that CCL19, SELL and VCAM-1 may promote inflammatory cell infiltration and nerve injury through synergistic effects in the pathology of Bell’s palsy, and although there is no clear signaling axis linking the three indirect crosstalk, their joint role in immune cell recruitment and vascular endothelial activation may be key to the onset and progression of the disease [25,26,27]. CCL19 acts as a chemokine that attracts T cells and dendritic cells to migrate toward inflammatory sites, whereas SELL, as an adhesion molecule, plays an important role in leukocyte rolling and adhesion, facilitating the penetration of immune cells through the vascular wall and into diseased tissues, which synergistically enhances local inflammatory responses with CCL19 [25,28]. In addition, this process may be further amplified by the up-regulation of VCAM-1 expression, which promotes strong binding of immune cells to endothelial cells by binding to VLA-4 and increases the permeability of the blood-nerve barrier, allowing more immune cells to penetrate into peripheral tissues of the facial nerve, thereby exacerbating local inflammation and nerve injury [26,29]. Thus, CCL19, SELL and VCAM-1 may not be sequential activation of a single pathway, but rather synergize through different mechanisms such as chemotaxis, adhesion and barrier modulation to drive the inflammatory response and pathological injury in Bell’s palsy. Targeted modulation of these molecules may help to reduce immune cell infiltration, alleviate inflammation, and improve neural repair, thus providing new directions for the treatment of Bell’s palsy.

Checking the related studies of other proteins, we found that IL-27RA, OSM and JAK/STAT signaling pathways may play a key role in the development of Bell’s palsy. Bell’s palsy is closely related to viral infections, particularly herpes simplex virus (HSV-1) infection [22]. After the virus enters the nervous system through the facial nerve, it triggers a neuroinflammatory response, leading to facial nerve damage. Studies have shown that viral infection activates the body’s immune response, prompting the release of inflammatory factors, which further exacerbates nerve damage [2,30]. The JAK/STAT pathway may act as a key signaling pathway in the immune response, regulating the activation of inflammatory cells and the onset of the immune response [31,32]. Upon viral infection, cells recognize and activate JAK proteins via receptors, which, in turn, activate STAT proteins, inducing the transcription of specific genes that are commonly associated with immune and inflammatory responses [33,34]. IL-27RA is the main receptor for IL-27 signal transduction, which can regulate T cell differentiation through the JAK/STAT cascade signaling pathway, and has a dual regulatory effect in maintaining immune homeostasis and mediating inflammatory responses [35,36]. MR analysis in this study showed that elevated IL-27RA expression was positively correlated with the risk of Bell’s palsy, suggesting that it may be involved in the disease process by regulating T cell-mediated inflammatory response. Previous studies have shown that IL-27RA may exert immunosuppressive effects by promoting the differentiation of Foxp3^+^ regulatory T cells (Tregs), and may also promote autoimmune inflammatory responses in specific microenvironments [37,38,39].

As an IL-6 family cytokine, OSM exerts its biological functions mainly through the JAK/STAT signaling pathway (especially STAT3-mediated signaling) [40,41]. During neurological inflammation, OSM induces CCL20 secretion and promotes Th17 cell recruitment, thereby disrupting the integrity of the blood–brain barrier and exacerbating neuroinflammation. MR analysis in the present study further confirmed the causal role of OSM in the pathogenesis of Bell’s palsy, suggesting that its aberrant expression may influence the disease process by promoting neuroinflammation [42]. Notably, the JAK/STAT signaling pathway, as a common downstream signaling axis of IL-27RA and OSM, has been shown to be aberrantly activated and closely associated with various inflammatory diseases and neuroimmune disorders [43,44].

Our results of the analyses suggest that HMOX2, DEFB128, DDX58 and ITM2A may act as risk factors for Bell’s palsy, but their association with the JAK/STAT signaling pathway is weak [45,46,47,48]. In addition, studies on these factors are more limited and no specific inhibitors or targeted intervention strategies have been identified against them. Although no study has clearly revealed the direct relationship between TNFRSF12A (TWEAK receptor) and the JAK/STAT pathway in inflammation, it has been demonstrated that TWEAK/Fn14 induces IFNβ and promotes apoptosis in tumor cells via the JAK-STAT pathway, and this process is JAK-dependent [49]. These results suggest that TNFRSF12A is closely related to the JAK/STAT pathway and may play an important role in the inflammatory response. In a study by Xiao et al., it was found that Ginkgo Flavonol Glycosides may attenuate ischemia–reperfusion injury in the heart and brain of myocardial ischemia–reperfusion injured mice by downregulating the TWEAK-Fn14 axis. In contrast, Ginkgolides may protect the brain from ischemia–reperfusion injury in cerebral ischemia–reperfusion injured mice by upregulating this signaling pathway, offering protective effects against ischemia–reperfusion injury in both the brain and heart [50]. We further found that the expression levels of JAK/STAT pathway-related effector molecules were positively correlated with the risk of Bell’s palsy, which genetically supports the key role of this pathway in disease development.

In our study, BLVRB and VEGF-A as potential protective factors in Bell’s palsy may contribute to the reduction in Bell’s palsy. However, VEGF-A has complex bi-directional regulatory properties: its protective effect promotes nerve repair mainly through VEGFR2-mediated neovascularisation and secretion of neurotrophic factors such as BDNF, but attention should also be paid to its potential risk of exacerbating inflammatory responses by increasing vascular permeability [51,52]. Currently, clinically available VEGF-A modulators include drugs such as cilostazol and metformin [53,54,55]. In contrast, BLVRB, which catalyzes the conversion of biliverdin to bilirubin via biliverdin reductase activity, can effectively scavenge reactive oxygen radicals and exert a unique antioxidant neuroprotective effect [56,57]. However, specific pharmacological interventions targeting this pathway are still in the developmental stage. Therefore, in the future, JAK-STAT pathway inhibitors (e.g., Tofacitinib, Baricitinib) may be a major research hotspot for the prevention and treatment of Bell’s palsy [58,59].

Although the present study systematically assessed the potential causal relationship between inflammatory immune-related proteins and Bell’s palsy based on Mendelian randomization analyses and conjugated bioinformatics validation. Firstly, the study was mainly based on data from European populations and lacked extensive validation in other ethnicities and regions, and the generalisability of its findings still needs to be further assessed. Secondly, as the cerebrospinal fluid (CSF) protein expression data of patients with Bell’s palsy could not be obtained, and the levels of inflammatory factors in the CSF may more directly reflect the central immune milieu of the disease, future studies should incorporate data from cerebrospinal fluid or facial nerve tissue samples for more in-depth investigation. In addition, although the MR method reduces the influence of confounding factors in traditional observational studies to a certain extent, it is still unable to completely circumvent potential gene–environment interactions and other systematic biases that may affect causal inference. Therefore, further experiments are needed to validate the pathogenesis of Bell’s palsy.

## 5. Conclusions

This study revealed a potential causal relationship between 70 inflammatory proteins, 77 metabolites and 26 immune cells with Bell’s palsy. After double validation, HMOX2, TNFRSF12A, DEFB128, ITM2A, DDX58, and BLVRB were identified as risk factors, while VEGF-A was considered a protective factor. Co-localization analysis further identified VCAM-1, CCL19, OSM and IL27RA as key risk proteins. The results suggest that the JAK/STAT signaling pathway may be a potential key target for intervention in Bell’s palsy, and its regulation may provide new directions and opportunities for therapeutic strategies and drug discovery in this disease.

## Figures and Tables

**Figure 1 biomedicines-13-00957-f001:**
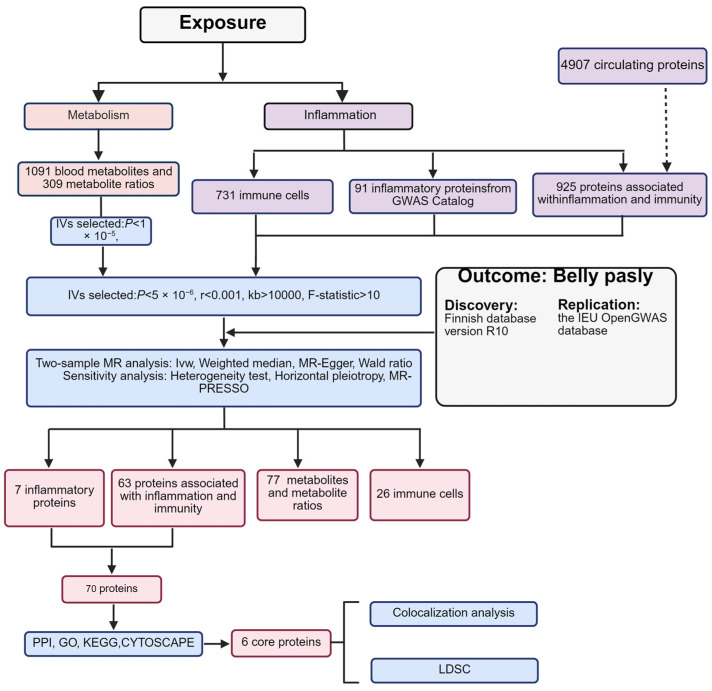
Main process steps of this experiment.

**Figure 2 biomedicines-13-00957-f002:**
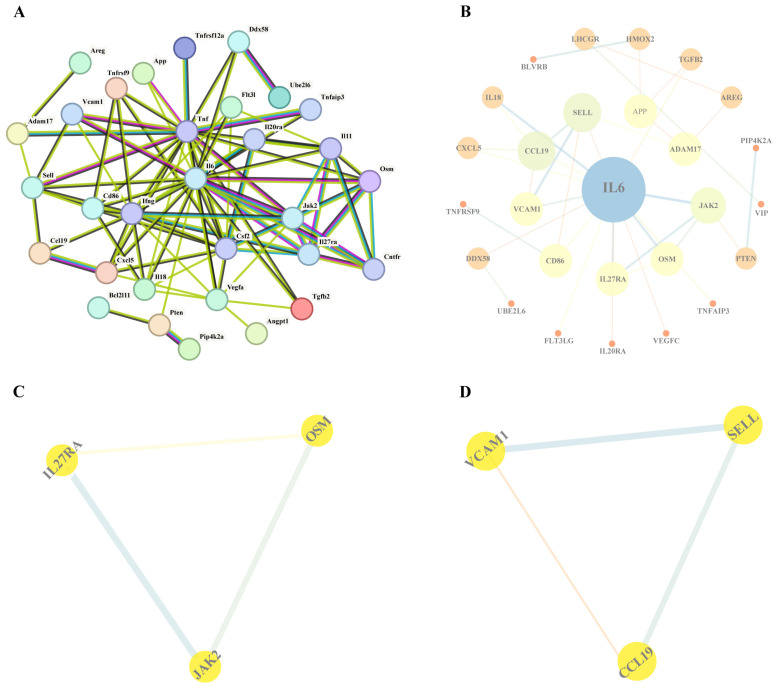
The results of the PPI network interaction and core protein selection are as follows. (**A**) The 31 proteins identified through the PPI network selection are as follows. (**B**) Further analysis of the 31 proteins was conducted using Cytoscape software (version 3.7.1). (**C**,**D**) Identification of two subnetworks comprising six hub proteins using the MCODE plugin.

**Figure 3 biomedicines-13-00957-f003:**
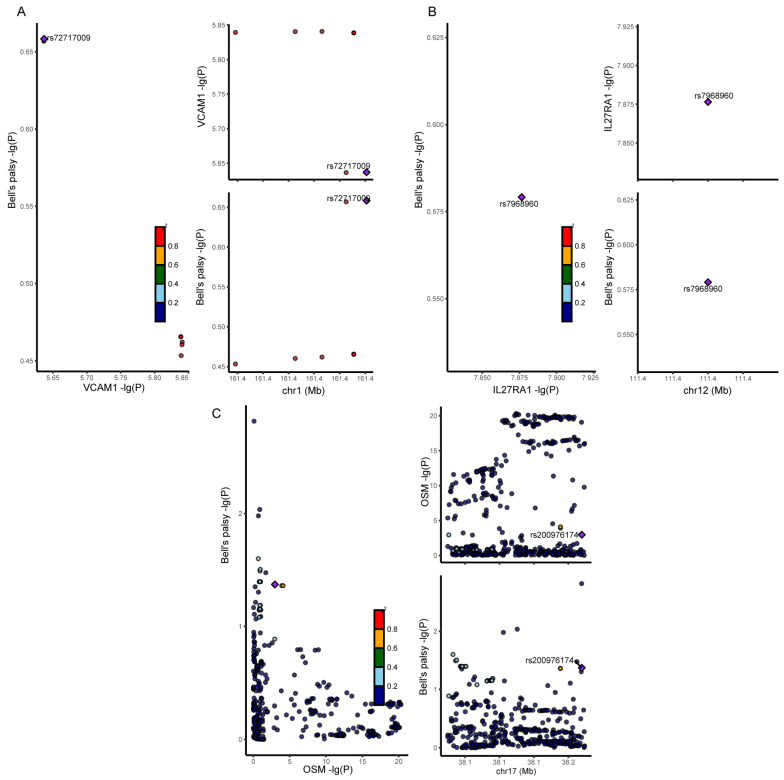
The results of colocalization analysis. (**A**) Colocalization analysis results between VCAM-1 and Bell’s palsy. (**B**) Colocalization analysis results between IL27RA and Bell’s palsy. (**C**) Colocalization analysis results between OSM and Bell’s palsy.

**Table 1 biomedicines-13-00957-t001:** Main MR results of core proteins and Bell’s palsy.

Exposure	Outcome	nSNP	OR (95%CI)	*p*-Val	Heterogeneity		Pleiotropy		MR Presso		LDSC		Color
					MR–Egger				Global Test				
					Q	Q_*p*-Val	Egger Intercept	*p*-Val	RSSobs	*p*-Val	rg	*p*-rg	PP.H4
CCL19	Bell palsy	23	1.15 (1.03, 1.29)	0.013	23.10148	0.339	−0.011963	0.146	39.73911	0.331	0.2105	0.042	0.0075119
SELL	Bell palsy	49	1.16 (1.06, 1.27)	<0.001	43.825816	0.605	0.010615	0.084	49.08359	0.548	0.0398	0.733	-
VCAM1	Bell palsy	27	1.23 (1.05, 1.43)	0.009	14.857993	0.945	0.0176444	0.162	18.20016	0.916	0.2478	0.017	0.0224088
IL27RA	Bell palsy	21	1.07 (1.01, 1.13)	0.031	10.554531	0.938	0.0175772	0.259	13.12543	0.926	0.0634	0.843	0.0213167
OSM	Bell palsy	15	1.17 (1.01, 1.36)	0.042	9.5880892	0.727	−0.008722	0.632	10.83632	0.803	0.41934	0.021	0.0117855
JAK2	Bell palsy	17	1.37 (1.04, 1.81)	0.023	17.758644	0.276	−0.006678	0.716	20.32988	0.351	−0.008	0.974	-

MR, Mendelian randomization; nSNP, single-nucleotide polymorphism; ORs, odds ratios; LDSC, linkage disequilibrium score correction; rg, genetic correlation.

## Data Availability

All data generated or analysed during this study are included in this published article (and its Appendix A files).

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
