# Peer review of "Exploring the Role of Inflammation and Metabolites in Bell’s Palsy and Potential Treatment Strategies"

_biomedicines, 2025, doi:10.3390/biomedicines13040957_

Round 1

Reviewer 1 Report

Comments and Suggestions for Authors

I reviewed the article titled "Exploring the role of inflammation and metabolites in Bell's palsy and potential treatment strategies" submitted by Lu et al. The article offers a thorough exploration of the genetic and molecular processes involved in Bell's palsy. The research work presented by authors employs a 2-sample Mendelian Randomization strategy based on the data from the Finnish database and IEU OpenGWAS. It is done to discover possible causal relations between Bell's palsy and inflammatory proteins, metabolites, and immune cell traits. The results identify 70 inflammatory proteins, 77 metabolites, and 26 immune cell characteristics as being potentially linked with Bell's palsy. The most significant validated proteins are BLVRB, HMOX2, TNFRSF12A, DEFB128, ITM2A, VEGF-A, and DDX58, and bioinformatics-based PPI network analysis indicates JAK2, IL27RA, OSM, CCL19, SELL, and VCAM-1 as central proteins. Bioinformatics analysis also indicates participation of cytokine-cytokine receptor interactions and immune regulation in Bell's palsy pathogenesis. Strengths are the strong genetic design, validation outside of European populations, and extensive bioinformatics analysis.
Nevertheless, limitations in the study do exist, for example, exclusive focus on the European populations, absence of data from cerebrospinal fluid, and inherent confounding issues not adequately eliminated. Generally speaking, this investigation gives valuable knowledge regarding molecular pathology of Bell's palsy as well as new therapeutic targets awaiting experimental validation. Please add the limitations of the study in discussion section. Please take care of the iThenticate report values also.

Author Response

Strengths are the strong genetic design, validation outside of European populations, and extensive bioinformatics analysis. Nevertheless, limitations in the study do exist, for example, exclusive focus on the European populations, absence of data from cerebrospinal fluid, and inherent confounding issues not adequately eliminated. Generally speaking, this investigation gives valuable knowledge regarding molecular pathology of Bell's palsy as well as new therapeutic targets awaiting experimental validation. Please add the limitations of the study in discussion section. Please take care of the iThenticate report values also.

         Response: We appreciate your question. We have followed your comments to add limitations in line  394-404. Additionally, we have carefully re-examined this article and made effort to minimize the iThenticate report values. Thank you again for your valuable comments.

Reviewer 2 Report

Comments and Suggestions for Authors

The authors submitted an article titled “Exploring the role of inflammation and metabolites in Bell's palsy and potential treatment strategies” that aims to explore the potential causal relationship between Bell's palsy and peripheral blood inflammatory proteins, metabolites, and immune cell characteristics.

The authors are encouraged to address the following concerns with the manuscript:

1) When analyzing the 4,907 plasma proteins, 731 immune cell traits, 91 inflammatory proteins, and 1,400 metabolites, did you have any demographic information for these patients that may be shown in a table to show the number of males versus females and the age groups? Please provide the demographics information to have a proper manuscript.

2) You identified 70 inflammatory proteins, 77 metabolites, and 26 immune cell traits as potentially causally associated with Bell's palsy. Please provide the demographic characteristics describing the patients from these samples. Age, Age-Group, Gender (Male versus Female), any medications the patients were taking.

3) For the results of the external validation that showed: BLVRB, HMOX2, TNFRSF12A, DEFB128, ITM2A, VEGF- 26 A, and DDX58 remaining significantly associated with the outcome variable, please provide a Decision Tree Diagram using these aforementioned genes to be able to use this information clinically. 

4) For each of these genes: BLVRB, HMOX2, TNFRSF12A, DEFB128, ITM2A, VEGF- 26 A, and DDX58, please provide information in a table if the tissues with the highest gene expression.

5) Please share if any existing pharmacological compounds can modulate the genes: BLVRB, HMOX2, TNFRSF12A, DEFB128, ITM2A, VEGF- 26 A, and DDX58 to help is potential drug repurposing.

6) In Line 427-428 you state: Importantly, our analysis provides potential new therapeutic targets for Bell's palsy. —> Please state which existing compounds already modulate these therapeutic targets.

7) Please discuss the gender differences known about diagnosis of Bell’s Palsy and potential mechanisms as to why more common in middle-aged women in relation to your findings.

8) Comment on the influence of estrogen on: BLVRB, HMOX2, TNFRSF12A, DEFB128, ITM2A, VEGF- 26 A, and DDX58.

9) Please comment on the influence of testosterone on: BLVRB, HMOX2, TNFRSF12A, DEFB128, ITM2A, VEGF- 26 A, and DDX58.

10) Please provide a list of medications and/or lifestyle of the patient’s samples represented in your dataset.

Author Response

1) When analyzing the 4,907 plasma proteins, 731 immune cell traits, 91 inflammatory proteins, and 1,400 metabolites, did you have any demographic information for these patients that may be shown in a table to show the number of males versus females and the age groups? Please provide the demographics information to have a proper manuscript.

Response: Thanks for your suggestion. We sincerely thank the reviewer for careful reading. The MR study is a secondary analysis based on published GWAS data. We collected demographic information from the original studies as comprehensively as possible and organized these details in Supplementary Table S34, making them available for readers' reference as needed.

2) You identified 70 inflammatory proteins, 77 metabolites, and 26 immune cell traits as potentially causally associated with Bell's palsy. Please provide the demographic characteristics describing the patients from these samples. Age, Age-Group, Gender (Male versus Female), any medications the patients were taking.

Response: Thanks for your suggestion. According to the characteristics of MR analysis, the exposure factors mentioned above that show causal associations with Bell's palsy do not possess independent demographic information but should share the same demographic characteristics as those in the original study. Additionally, as previously mentioned, MR studies are secondary analyses based on published GWAS data, and we did not identify information regarding medications currently taken by participants in the original research.

3) For the results of the external validation that showed: BLVRB, HMOX2, TNFRSF12A, DEFB128, ITM2A, VEGF- 26 A, and DDX58 remaining significantly associated with the outcome variable, please provide a Decision Tree Diagram using these aforementioned genes to be able to use this information clinically.

Response: We appreciate your question. Clinical interventions for genes such as BLVRB, HMOX2, TNFRSF12A, DEFB128, ITM2A, VEGF-A, and DDX58 have not yet been established, and existing guidelines do not provide specific recommendations. Therefore, constructing a rigorous and well-defined clinical decision tree diagram based on the available data is not currently feasible. However, future studies may help clarify the role of these genes in Bell’s palsy and their potential as therapeutic targets.

4) For each of these genes: BLVRB, HMOX2, TNFRSF12A, DEFB128, ITM2A, VEGF- 26 A, and DDX58, please provide information in a table if the tissues with the highest gene expression.

Response: We appreciate your question. We used the MR analysis of the data related to blood tissue in the database, in which the data could not show the gene expression. The results of the MR analysis were mainly related to Beta value, P value.

Gene

Beta

P (discovery cohort)

P(validation cohort)

BLVRB

-0.342491788

0.039475211

0.029994793

HMOX2

0.292083604

0.037291204

0.01196231

TNFRSF12A

0.685684181

0.002875031

0.04777811

DEFB128

0.232302564

0.036943551

0.00854748

ITM2A

0.257490443

0.012687979

0.003962844

DDX58

0.297000532

0.036334868

0.015427251

VEGF-A

-0.112135115

0.006855811

0.032221187

5) Please share if any existing pharmacological compounds can modulate the genes: BLVRB, HMOX2, TNFRSF12A, DEFB128, ITM2A, VEGF-A, and DDX58 to help is potential drug repurposing.

Response: We appreciate your question. As discussed in the manuscript, based on the current literature, no well-defined pharmacological interventions have been identified for HMOX2, DEFB128, ITM2A, and DDX58. However, VEGF-A and TNFRSF12A are associated with the JAK/STAT signaling pathway. Regulatory drugs targeting vegf-a and TNFRSF12A have been studied accordingly, suggesting to us that they have potential regulatory possibilities.In addition, several modulators of this signalling pathway are in clinical use. These include JAK inhibitors such as tofacitib and baricitinib, which may be of potential interest in modulating these targets.

6) In Line 427-428 you state: Importantly, our analysis provides potential new therapeutic targets for Bell's palsy. > Please state which existing compounds already modulate these therapeutic targets.

Response : Thank you to the reviewers for their questions. We have refreshed the presentation of the Discussion section and analysed the potential targets of intervention for the positive proteins and the links between them. In conjunction with the answer to the sixth question, we focused our analysis in the Discussion to conclude that the JAK/STAT pathway, Vegf-a, and TNFRSF12A may be potential targets for intervention in future Bell's palsy.

7) Please discuss the gender differences known about diagnosis of Bells Palsy and potential mechanisms as to why more common in middle-aged women in relation to your findings.

Response: Thanks for your suggestion. There is currently no high-quality evidence confirming gender as a risk factor for Bell’s Palsy. However, some studies have suggested that pregnancy may be a potential risk factor for Bell’s Palsy in female patients.[1; 2] Nevertheless, from an epidemiological perspective, pregnancy has not been widely recognized as a significant contributor to an increased incidence of Bell’s Palsy.[3; 4] Thus, we generalise in our discussion of line 384-387.

8) Comment on the influence of estrogen on: BLVRB, HMOX2, TNFRSF12A, DEFB128, ITM2A, VEGF- 26 A, and DDX58.

Response: We appreciate your question. Estrogen plays an important role in the regulation of several genes, especially VEGF-A, which has a key role in angiogenesis and vascular function. Many studies have shown that estrogen can up-regulate the expression of VEGF-A, promote angiogenesis and maintain vascular integrity.[5; 6] However, considering the dual effects of vegf-a on inflammation,[7; 8] whether estrogen can be a regulator of vgef-a needs further study. The remaining proteins have not been reported in the literature. Thus, we generalise in our discussion of line 384-387

9) Please comment on the influence of testosterone on: BLVRB, HMOX2, TNFRSF12A, DEFB128, ITM2A, VEGF- 26 A, and DDX58.

Response: We appreciate your question. Testosterone can regulate VEGF-A expression through the androgen receptor (AR) pathway, but with different outcomes in different situations. In androgen-sensitive tissues such as prostate and testis, testosterone induces VEGF-A expression to support angiogenesis and organ development.[9] However, in certain prostate diseases and cancer studies, testosterone may accelerate pathological processes by promoting aberrant angiogenesis through the VEGF-A pathway. [10; 11] Its feasibility, although showing potential for VEGF-A regulation, requires further experimental authentication. The other positive proteins did not have high-quality studies demonstrating their relationship with androgens. Thus, we generalise in our discussion of line 384-387

10) Please provide a list of medications and/or lifestyle of the patient’s samples represented in your dataset.

Response: Thank you for your suggestion. As previously mentioned, Mendelian Randomization (MR) studies are secondary analyses based on published Genome-Wide Association Studies (GWAS) data. Unfortunately, the original studies did not provide specific information regarding the participants' medication usage or lifestyle factors, so we were unable to include these details in our dataset.

[1] L. Lansing, S.B. Wendel, M. Hultcrantz, and E. Marsk, Bell's Palsy in Pregnancy and Postpartum: A Retrospective Case-Control Study of 182 Patients. Otolaryngol Head Neck Surg 168 (2023) 1025-1033.

[2] N.N. Carmel Neiderman, Y. Netanyahu, O.J. Ungar, O. Handzel, R. Masarwy, R. Abu-Eta, L. Reicher, and Y. Oron, Bell's palsy and pregnancy: Incidence, comorbidities and complications. A meta-analysis and systematic review of the literature. Clin Otolaryngol 48 (2023) 576-586.

[3] H. Jones, J. Hintze, F. Slattery, and A. Gendre, Bell's palsy in pregnancy: A scoping review of risk factors, treatment and outcomes. Laryngoscope Investig Otolaryngol 8 (2023) 1376-1383.

[4] J.T. Vrabec, B. Isaacson, and J.W. Van Hook, Bell's palsy and pregnancy. Otolaryngol Head Neck Surg 137 (2007) 858-61.

[5] X. Xiao, J. Liu, and M. Sheng, Synergistic effect of estrogen and VEGF on the proliferation of hemangioma vascular endothelial cells. J Pediatr Surg 39 (2004) 1107-10.

[6] G.M. Rubanyi, A. Johns, and K. Kauser, Effect of estrogen on endothelial function and angiogenesis. Vascul Pharmacol 38 (2002) 89-98.

[7] S. Wiszniak, and Q. Schwarz, Exploring the Intracrine Functions of VEGF-A. Biomolecules 11 (2021).

[8] A. Adini, V.H. Ko, M. Puder, S.M. Louie, C.F. Kim, J. Baron, and B.D. Matthews, PR1P, a VEGF-stabilizing peptide, reduces injury and inflammation in acute lung injury and ulcerative colitis animal models. Front Immunol 14 (2023) 1168676.

[9] L. Chodari, M. Mohammadi, V. Ghorbanzadeh, H. Dariushnejad, and G. Mohaddes, Testosterone and Voluntary Exercise Promote Angiogenesis in Hearts of Rats with Diabetes by Enhancing Expression of VEGF-A and SDF-1a. Can J Diabetes 40 (2016) 436-441.

[10] L. Trujillo-Rojas, J.M. Fernández-Novell, O. Blanco-Prieto, T. Rigau, M.M. Rivera Del Álamo, and J.E. Rodríguez-Gil, The onset of age-related benign prostatic hyperplasia is concomitant with increased serum and prostatic expression of VEGF in rats: Potential role of VEGF as a marker for early prostatic alterations. Theriogenology 183 (2022) 69-78.

[11] L. Cheng, S. Zhang, C.J. Sweeney, C. Kao, T.A. Gardner, and J.N. Eble, Androgen withdrawal inhibits tumor growth and is associated with decrease in angiogenesis and VEGF expression in androgen-independent CWR22Rv1 human prostate cancer model. Anticancer Res 24 (2004) 2135-40.

Reviewer 3 Report

Comments and Suggestions for Authors

Figures 2-3-4-5 are illegible. Can you think to put some of them in supplementary files?

Line 321: when you write about viral infection, can you add more details?

The discussion is so bundled that it is difficult to understand a "take home message"!

Can you re-write it, by inserting a table or some subsections for the more significant proteins?

The conclusions are so poor. Can you re-write them?

Author Response

1) Figures 2-3-4-5 are illegible. Can you think to put some of them in supplementary files?

Response: We sincerely appreciate your valuable comment. Initially, we believed including these figures in the main text would provide more intuitive demonstration of the results. In this revision, we have made every effort to enhance the clarity of Figures 2-3-4-5 and have simultaneously uploaded them as separate figure files. Should you still consider these figures unnecessary in the main text, we would be more than willing to make further revisions and relocate them to the Supplementary Files.

Line 321: when you write about viral infection, can you add more details?

Response: Thank for your suggestion. We have added more detailed information to enhance the explanation. (line 330-341).

The discussion is so bundled that it is difficult to understand a "take home message"!Can you re-write it, by inserting a table or some subsections for the more significant proteins?

Response: We sincerely thank the reviewer for careful reading. We restructured the Discussion to improve clarity, revising the unclear sections, focusing on the key findings and highlighting the role of the JAK/STAT signaling pathway in Bell's palsy, along with its potential for therapeutic intervention.

The conclusions are so poor. Can you re-write them?

Response: We sincerely appreciate the valuable comment. We re-write the Conclusion section to more clearly present the findings of this study.

Round 2

Reviewer 3 Report

Comments and Suggestions for Authors

I suggest to put figures 2 and 4 in a supplementary files to make them really readable and useful.

I accept the other revisions. 

Author Response

Commemts: “I suggest to put figures 2 and 4 in a supplementary files to make them really readable and useful.”
Response: Thanks for your suggestion. I have revised the manuscript according to the reviewer’s suggestion and moved Figures 2 and 4 to the supplementary files to enhance their readability and usefulness.